# Effect of a NICU to Home Physical Therapy Intervention on White Matter Trajectories, Motor Skills, and Problem-Solving Skills of Infants Born Very Preterm: A Case Series

**DOI:** 10.3390/jpm12122024

**Published:** 2022-12-07

**Authors:** Christiana Dodd Butera, Claire Rhee, Claire E. Kelly, Thijs Dhollander, Deanne K. Thompson, Jessica Wisnowski, Rebecca M. Molinini, Barbara Sargent, Natasha Lepore, Greg Vorona, Dave Bessom, Mary S. Shall, Jennifer Burnsed, Richard D. Stevenson, Shaaron Brown, Amy Harper, Karen D. Hendricks-Muñoz, Stacey C. Dusing

**Affiliations:** 1Division of Biokinesiology and Physical Therapy, Herman Ostrow School of Dentistry, University of Southern California, Los Angeles, CA 90033, USA; 2Victorian Infant Brain Studies and Developmental Imaging, Murdoch Children’s Research Institute, Parkville, VIC 3052, Australia; 3Turner Institute for Brain and Mental Health, School of Psychological Sciences, Monash University, Melbourne, VIC 3000, Australia; 4Developmental Imaging, Murdoch Children’s Research Institute, Parkville, VIC 3052, Australia; 5Department of Paediatrics, University of Melbourne, Parkville, VIC 3052, Australia; 6Departments of Radiology and Pediatrics (Neonatology), Children’s Hospital Los Angeles, Los Angeles, CA 90027, USA; 7Department of Physical Therapy, Virginia Commonwealth University, Richmond, VA 23284, USA; 8CIBORG Laboratory, Department of Radiology, University of Southern California, Los Angeles, CA 90089, USA; 9Departments of Pediatrics and Biomedical Engineering, University of Southern California, Los Angeles, CA 90089, USA; 10Department of Radiology, Virginia Commonwealth University, Richmond, VA 23284, USA; 11Department of Radiology, Children’s Hospital of Richmond at VCU, Richmond, VA 23284, USA; 12Division of Neonatology, Departments of Pediatrics and Neurology, University of Virginia, Charlottesville, VA 22903, USA; 13Division of Neurodevelopmental and Behavioral Pediatrics, Department of Pediatrics, University of Virginia School of Medicine, Charlottesville, VA 22903, USA; 14Department of Neurology, Virginia Commonwealth University, Richmond, VA 23284, USA; 15Department of Pediatrics, Virginia Commonwealth University School of Medicine, Children’s Hospital of Richmond at VCU, Richmond, VA 23284, USA

**Keywords:** early intervention, preterm infants, fixel-based analysis, corticospinal tract

## Abstract

Infants born very preterm (VPT; ≤29 weeks of gestation) are at high risk of developmental disabilities and abnormalities in neural white matter characteristics. Early physical therapy interventions such as Supporting Play Exploration and Early Development Intervention (SPEEDI2) are associated with improvements in developmental outcomes. Six VPT infants were enrolled in a randomised clinical trial of SPEEDI2 during the transition from the neonatal intensive care unit to home over four time points. Magnetic resonance imaging scans and fixel-based analysis were performed, and fibre density (FD), fibre cross-section (FC), and fibre density and cross-section values (FDC) were computed. Changes in white matter microstructure and macrostructure were positively correlated with cognitive, motor, and motor-based problem solving over time on developmental assessments. In all infants, the greatest increase in FD, FC, and FDC occurred between Visit 1 and 2 (mean chronological age: 2.68–6.22 months), suggesting that this is a potential window of time to optimally support adaptive development. Results warrant further studies with larger groups to formally compare the impact of intervention and disparity on neurodevelopmental outcomes in infants born VPT.

## 1. Introduction

Infants born very preterm (VPT, <32 weeks of gestation) are at high risk of developmental disabilities including cerebral palsy, and motor and cognitive delays [1,2]. During the first months of life, common impairments include difficulties with postural control, movement variability, visual motor skills, and motor learning [3,4,5]. Early brain injuries, e.g., periventricular leukomalacia (PVL) and intraventricular haemorrhage (IVH), are seen at higher rates in infants born VPT [6] and are predictive of a later diagnosis of cerebral palsy (CP) [7]. Infants born VPT also demonstrate greater subtle regional differences in neural white matter microstructure on magnetic resonance imaging (MRI) compared to term-born infants [8,9,10,11,12,13,14,15,16]. These differences have been associated with later cognitive and motor delays [17,18,19,20]. However, early identification of brain injury and white matter alterations is lacking and is a crucial prerequisite to effectively implement early interventions [21,22] during the first months of life, a window of rapid neuroplasticity.

Evidence indicates that early interventions such as Supporting Play Exploration and Early Development Intervention (SPEEDI2) are associated with improvements in developmental outcomes in infants born VPT [23,24]. However, no MRI studies have explored how early intervention impacts the brain over time in this population. In older children with brain injury and cerebral palsy, corticospinal tract reorganisation was observed after receiving constraint-induced movement therapy [25] and Hand and Arm Bimanual Intensive Therapy Including Lower Extremities [26]. Additionally, MRI studies of preterm infant cohorts are primarily cross-sectional and performed at term-equivalent age [27]. Longitudinal data collection at multiple time points from birth through the first year of life are crucial for assessing brain plasticity and recovery in relation to early intervention. Further research is needed in infants born VPT to assess intervention-induced changes in neurodevelopmental outcomes and their underlying neural substrates.

Brain imaging may play a vital role by providing insights into personalised medicine, to inform who will respond best to specific types of therapy, and when they are most effectively applied. The majority of previous work has used the diffusion tensor imaging (DTI) model to explore the relationships between white matter microstructure and neurodevelopmental outcomes [28,29,30,31,32,33,34,35]. However, DTI is inaccurate in its inability to resolve crossing fibre architectures, causing issues in identifying differences in white matter microstructure in many major white matter tracts. Recent advances in diffusion MRI analysis, including fixel-based analysis [36,37] and constrained spherical deconvolution (CSD) techniques [38,39,40], have resolved this crossing fibre challenge. Fixel-based analysis (FBA) is a framework for statistical analysis of fibre-specific metrics, including fibre density (FD), fibre bundle cross-section (FC), and combined fibre density and bundle cross-section (FDC) in each fibre population in a voxel (referred to as a “fixel”) [37]. A few studies have demonstrated that these fibre-specific measures can be used to detect differences in white matter microstructure between infants born preterm and those born at term [13], and that these differences may be associated with cognitive and motor outcomes at 1 and 2 years [20] and later cerebral palsy diagnosis [41]. However, it is still unknown how intervention can impact these outcomes, and if additional play-based measures of motor problem solving are related to these metrics. 

In this case, a series of six infants born VPT, we aim to (1) investigate whether changes in FD, FC, and FDC in regions associated with motor learning [left and right corticospinal tract; left and right cerebellar peduncles; subregions of the corpus callosum] are associated with motor, cognitive, and motor-based problem-solving outcomes and (2) if early intervention can impact these relationships at multiple timepoints over the first year of life. A case series affords an opportunity to examine how FD, FC, and FDC are impacted in infants receiving a daily physical therapy intervention in the first months of life (specifically SPEEDI2), and to examine other individual factors (race, ethnicity, socioeconomic status) that may be important to consider in future studies.

## 2. Materials and Methods

### 2.1. Participants

All eligible infants born VPT enrolled in the “Does Timing Matter? Supporting Play, Exploration, and Early Development Intervention” trial (SPEEDI2—ClinicalTrials.gov Identifier: NCT03518736) in early 2020 were approached for enrolment in the optional MRI substudy until late 2021. Eligible infants received care in one of the three participating neonatal intensive care units (NICUs), lived within 100 miles of the participating hospitals, were born <29 weeks of gestation, and were medically stable and off ventilator support between 35 and 42 weeks of gestation when the baseline developmental assessment was complete. Exclusion criteria included: non-English-speaking families or a diagnosis of genetic abnormality at the time of enrolment. Additional information on selection, allocation concealment and randomisation is available in the protocol paper for the primary SPEEDI2 intervention trial [42]. This study was approved by the Human Subjects Board at Virginia Commonwealth University and a caregiver provided consent for their child’s participation in the primary trial, supplemental imaging study, and access to their child’s medical records.

Six infants were enrolled in this supplemental study which added non-sedated MRI scanning to a 3-arm clinical trial (usual care, SPEEDI2 early starting in the NICU, SPEEDI2 late starting at 15 weeks post-baseline). The MRI scans were completed at four time points. Visit 1 consisted of a) a baseline assessment of developmental and neurological function completed in the NICU as soon as the infant was medically stable and b) an MRI completed within 72 h after NICU discharge. In effect, ‘Visit 1’ involved 2 visits, which were carried out to ensure the same outpatient scanner was used for all MRIs. The remaining visits were scheduled based on the time from the baseline: 15 weeks post-baseline (Visit 2), 30 weeks post-baseline (Visit 3), and 12 months post-baseline (Visit 4). With the exception of the first MRI, developmental outcomes were assessed on the same day. Six infants completed MRI within 72 h of NICU discharge (Visit 1), 5 infants completed MRI at Visit 2 (one missed due to COVID-19), 5 infants completed MRI at Visit 3 (one parent opted out of further MRI to avoid removing new earrings) and 4 infants completed MRI at Visit 4 (one infant did not remain asleep). Enrolled infants were randomised into one of 3 groups: usual care (Infants 3 and 4), SPEEDI2 early (Infants 2 and 6), SPEEDI2 late (Infants 1 and 5). Infant 5’s family opted out of intervention before starting, so this infant is presented with the usual care group, and SPEEDI2 early (*n* = 2) and SPEEDI2 late (*n* = 1) groups are collapsed and presented as the SPEEDI2 intervention group. 

### 2.2. Intervention

The Supporting Play Early Exploration and Developmental Intervention (SPEEDI2) aims to provide increased learning opportunities by engaging caregivers to provide an enriched environment for their infants born VPT. SPEEDI2 supports the motor and cognitive development through targeted environmental enrichment, sensory-motor learning opportunities, and collaboration between the parent, therapist, and infant to determine the best time and way to interact. Phase 1 consists of 5 sessions over 3 weeks, and focuses on supporting parents in learning about the intervention, how to identify the best times to provide learning opportunities, and how to balance the motor, cognitive, and social skill practice for their infant who may tire quickly, have ongoing state regulation issues, and have altered parent–child interactions after prolonged NICU admission. Phase 2 consists of 20 min daily parent-provided play activities that target increased cognitive and motor development over 3 months, as well as 5 in-home or clinic-based sessions in which the therapist and parent collaborate to determine the infant’s ideal level of developmental practice. Activities included in the sessions and parent activity booklet encourage skills such as head control, reaching, kicking, and tummy time, with self-directed movement, movement variability, object interaction, and social interaction. The primary intervention trial included three groups: SPEEDI2 Early, SPEEDI2 Late, and usual care. All groups received standard care in the NICU and community. The SPEEDI2 Early group received intervention starting in the NICU, and the SPEEDI2 Late intervention began immediately after assessment Visit 2 and both interventions were for 15 weeks. Further details are available in the SPEEDI2 protocol paper [42].

### 2.3. MRI Administration & Protocol

Non-sedated MRI scans were performed in a 20-channel coil using a 3 T Siemens Skyra scanner in the outpatient department of Children’s Hospital of Richmond (CHOR). To increase the likelihood of natural sleep during the MRI, scans were completed in the evening, and infants were provided with earmuffs, fed, and rocked to sleep. To encourage sustained sleep, decreased movement, and midline head alignment, infants were swaddled in a papoose and foam blocks were placed around the head. Magnetization Prepared Rapid Gradient Echo, 3D SPACE (Sampling Perfection with Application optimised Contrasts using different flip angle Evolution), and diffusion-weighted echo-planar images (40 gradient directions at b = 1000 s/mm^2^ and 1 b = 0 image) were acquired. Total scan time varied between 18–30 min depending on data quality and movement artifacts.

### 2.4. Descriptive and Outcome Measures

Outcome measures from the primary study of SPEEDI2 were completed at Visit 1 between 35–42 weeks of gestation as soon as the infant was medically stable, at Visit 2 and 3 at 15 and 30 weeks post-baseline, and Visit 4 12 months later. The primary study includes a visit at 24 months of chronological age which is not included in this analysis.

Prechtl’s General Movement Assessment (GMA) was used at Visit 1 and Visit 2 to characterise the infants’ general movements, a type of spontaneous movement that has been shown to be a reliable tool for determining whether infants are at high risk of cerebral palsy, specifically at 3 to 5 months of age [43]. At Visit 1, each infant’s writhing general movements were classified as normal or abnormal (poor-repertoire, cramped synchronised, or chaotic). At Visit 2, each infant’s fidgety general movements were classified as normal, abnormal or absent fidgety [44]. 

The Test of Infant Motor Performance (TIMP) is a standardised instrument for assessing neuromotor development of infants 34 weeks postmenstrual age to 4 months of corrected age [45] and is a highly recommended Common Data Element (CDE) [46] in infants with or at risk of cerebral palsy. The TIMP is reliable and sensitive to change with increasing age and skill [45] and has excellent discriminative, evaluative, and predictive reliability in infants born preterm [47]. The TIMP was used at Visit 1 and Visit 2 to measure a change in the motor domain.

The Bayley Scales of Infant and Toddler Development, 3rd edition (Bayley-III) is a norm-referenced test which assesses developmental domains, including cognition and motor, of infants from 3 to 42 months of age [48]. The Bayley-III test–retest reliability for each subtest domain is between 67 and 86 and demonstrates good validity [48]. The Bayley-III is a highly recommended CDE for neurocognitive and developmental assessment. The Bayley-III was used at Visits 2, 3, and 4 to measure a change in the cognitive and motor domains. Bayley-III outcomes reported here include both a raw score and a composite score. The raw scores measure a change in total scores on test items, and the composite score is standardised to allow for comparison of an individual’s score when normalised with an age-matched typical sample.

The Assessment of Problem Solving in Play (APSP-4) is a play-based measure of early problem-solving skills and has been used with children with motor impairments and is correlated with Bayley-III cognitive raw scores. The APSP-4 is administered in 6 min, presenting the same set of three toys, each for two minutes and quantifying the most advanced problem-solving skills in each of 90, 4 s blocks. Assessors interact with the child only if the child initiates but do not provide any hints or insights into how to solve the toy. The assessment was completed in sitting with the assessor providing the least amount of postural support the child needed to maintain upright sitting. All assessments were completed in the home or in a home-like setting in a research lab, and two were completed virtually from home due to COVID-19. The frequency of each problem-solving skill was then entered into a weighted scoring model and the summed score was divided by assessment length to produce a single rate-per-minute problem-solving score. Twenty percent of all videos were double-scored for inter-rater (ICC = 0.98, 95% CI = (0.96, 0.99)) and intra-rater (Agreed/Agree + Disagree × 100 = 94) reliability. 

NIH CRF Common Data Elements CP Imaging. All available MRI images (20 images) were reviewed and scored using the Neuroimaging Cerebral Palsy Magnetic Resonance Imaging Case Report Form [49] by an expert neuroscientist with more than a decade of experience in MRI of perinatal brain injury. Sections in the form included General Information, Overall Assessment, Signal and Structural Abnormalities (e.g., lesions), Brain Abnormalities (e.g., haemorrhage, oedema, cerebral atrophy, ventricular dilation, assessment of commissures), and Summary Classifications. The summary classification scores were used for analysis, specifically white matter injury (minimal, moderate, severe, or N/A) and pattern of injury (0, 1A, 1B, 2A, 2B, 3, or N/A). White matter injury classification was determined based on the number of punctate lesions with minimal being less than or equal to three punctate lesions, moderate being greater than three punctate lesions, and severe being too many to count. The pattern of injury was determined based on the location and extent of brain injuries: 0 as normal; 1A as minimal cerebral lesions not affecting the basal ganglia (BG), thalamus (T), anterior limb of the internal capsule (ALIC), or posterior limb of the internal capsule (PLIC); 1B as extensive cerebral lesions not affecting the BG, T, ALIC, or PLIC; 2A as only involvement of the BG, T, ALIC, or PLIC, or watershed infarction; 2B as the involvement of the BG, T, ALIC, PLIC, and any other cerebral lesions; 3 as cerebral hemispheric devastation; or N/A as not applicable. 

Diffusion MRI-derived Metrics. FD, FC and FDC were calculated for white matter tracts of interest: (i) left and right corticospinal tract (CST), (ii) left and right cerebellar peduncles (inferior, middle, and superior), and (iii) seven subregions of the corpus callosum. Each tract was chosen based on previous studies showing alterations in infants born preterm, and their importance in voluntary movement, motor learning, and motor processing.

### 2.5. Statistical Data Analysis

Diffusion MRI-derived metrics (FD, FC, FDC). Blinded team members performed typical state-of-the-art fixel-based analysis pipeline steps [36,37], including denoising [50], Gibbs-ringing correction [51], motion correction [52], estimation of 3-tissue response functions and averaging response functions across the images from the fourth (oldest) timepoint [53], upsampling to 1.5 mm isotropic voxels, brain mask estimation [54], Single-Shell 3-Tissue Constrained Spherical Deconvolution [40], and intensity normalisation (global and bias fields) [55]. These steps resulted in white matter fibre orientation distributions (FODs) and grey matter and CSF compartments for each participant. A study-specific white matter FOD template was constructed, and all infants’ FOD images were registered and warped to this template [56]. In template space, fixels were segmented and reoriented, correspondence with common template fixels was established, and fixel-wise fibre density (FD), fibre cross-section (FC, in log form) and fibre density and cross-section (FDC) metrics were computed for each infant, as previously described [36]. White matter tracts were segmented in template space using TractSeg [57], following which FD, FC and FDC values for each infant were averaged across each tract, plotted, and a Hedges g calculated. Note these metrics are entirely tract-specific, and not influenced by those of other crossing white matter tracts.

General. Descriptive statistics were computed in IBM SPSS Statistics Edition 28 and were used to describe the demographic details of the participants. Within and across all infants, Pearson correlation and paired samples *t*-tests were used to analyse change over time. Due to the case series design and small sample size, a complete statistical group analysis was beyond the scope of this project. Qualitative and descriptive changes for developmental measures, FD, FC and FDC, and brain injury classification at Visit 1, Visit 2, Visit 3, and Visit 4 were reported. To explore the role of intervention on neuroplastic changes, FD, FC, and FDC, and brain injury classification for all infants who had at least a baseline and 12 months visit from each group (SPEEDI2 intervention [collapsed] and usual care), were visually represented using line graphs and descriptively analysed.

## 3. Results

### 3.1. Intervention Compliance

The three infants in the intervention each completed the 10 parent-therapy collaborative visits with a mean session duration of 55 ± 18 min. All parents reported they were adjusting well to parenting and were using the intervention approaches as requested.

### 3.2. Case Descriptions: Developmental Assessments

Figure 1 shows the outcome measures over time for infants receiving intervention and usual care.

#### 3.2.1. Usual Care Group

Infant 3: Infant 3 was a white biological male born at 27 weeks with a birth weight of 1030 g and Appearance, Pulse, Grimace, Activity, and Respiration (APGAR) scores of 3 at 1 min and 6 at 5 min. This infant was in the NICU for 87 days with a neonatal medical index (NMI) of 5. This infant falls in the minimal brain injury classification category (1A) and was randomised to the usual care group (not receiving intervention). The mother reported having a professional school degree for herself, a Bachelor’s degree for the father, and a household income category of USD 100,000 and over (Table 1). Both parents were employed full-time. The infant was discharged home with at least one parent and a sibling. At NICU discharge the infant had a normal GMA and no motor delays on TIMP scores. Bayley-III scores at Visit 2 (the first eligible visit for this measure for all infants) indicate low average motor performance (85 composite, or 1 SD below the mean) and average cognitive performance (95 composite; Table 2).

This infant’s TIMP z-score went down by over half a standard deviation (−0.650) from Visit 1 to Visit 2, with Visit 2 score suggesting a risk of long-term motor impairments. Over time, this infant improved steadily in their raw scores for cognitive and motor skills. At Visit 4, Infant 3 had a 91 composite for motor skills and 115 composite for cognitive skills on the Bayley-III. The highest motor composite score for Infant 3 was seen at Visit 3 (97 composite). This infant was the only one who had a positive upward trend from Visit 2 to Visit 4 on the cognitive composite scores. Cognitive composite scores in all other infants decreased from Visit 2 to Visit 4, although some increased between Visits 2 and 3 or between Visits 3 and 4. Infant 3’s global problem-solving scores on the APSP steadily increased from Visit 2 to Visit 4 (Figure 1). 

Infant 4: Infant 4 was a black biological female born at 27 weeks with a birth weight of 970 g and APGAR scores of 7 at 1 min and 8 at 5 min. This infant was in the NICU for 66 days with an NMI of 3. This infant was without brain injury and was randomised to the usual care group. The mother reported completing some college for herself, some high school for the father, and a household income category of USD 35,000 to USD 49,999 (Table 1). The mother reported that the father was working but not full-time. The infant was discharged home with at least one parent and sibling. At NICU discharge this infant had a normal GMA, and no motor delays were demonstrated on TIMP scores. This infant’s TIMP z-score increased (0.227) from Visit 1 to Visit 2, and scores at both visits were in the normal range. Bayley-III scores at Visit 2 indicated superior motor performance (121 composite) and average cognitive performance (95 composite; Table 2). 

Over time, this infant improved in their raw scores for cognitive and motor skills. For motor performance, Bayley-III composite scores dropped drastically to 76 composite at Visits 3 and 4. Cognitive performance dropped as well to 85 composite at Visit 4. Global problem-solving scores on the APSP increased from Visit 2 to Visit 4, however, this infant’s slope of change from Visit 3 to Visit 4 was much flatter than other infants, and this infant had the lowest global APSP score in the group at Visit 4 (Figure 1).

Infant 5: Infant 5 was a biological female born at 27 weeks with a birth weight of 1000 g and APGAR scores of 5 at 1 min and 6 at 5 min. This infant was in the NICU for 57 days with an NMI of 3. This infant was without brain injury and was initially randomised to the late intervention group, but opted out of intervention visits. This infant is considered part of the usual care group in this analysis since they never received an intervention. Infant’s race, maternal and paternal education levels and employment statuses, household income, infant location post-discharge, and people living with the infant were not reported (Table 1). At NICU discharge this infant had a normal GMA, and no motor delays were demonstrated based on TIMP scores. Bayley-III scores at Visit 2 indicate average motor performance (107 composite) and average cognitive performance (90 composite, Table 2).

This infant’s z-score, went down by greater than 1 standard deviation (−1.164) from Visit 1 to Visit 2, but neither visit qualified as motor delayed. Over time, this infant improved in their raw scores for cognitive and motor skills. The cognitive raw scores had the steepest increase among all the infants; however, cognitive composite scores decreased from 90 composite at Visit 2 to 85 composite at Visit 4. Interestingly, there was a positive increase in cognitive composite scores in this infant from Visit 2 to Visit 3, where the infant’s score increased from 90 composite to 105 composite (during the same time period they were receiving SPEEDI2 intervention). Motor composite scores dropped consistently over time and were at 94 composite at Visit 4. This infant had the lowest global problem-solving scores in the group on the APSP in both Visit 2 and Visit 3. Global problem-solving scores increased from Visit 2 to Visit 3, and this infant did not complete the APSP at Visit 4 (Figure 1).

#### 3.2.2. SPEEDI2 Intervention Group

Infant 1: Infant 1 was a black biological male born at 23 weeks with a birth weight of 580 g and APGAR scores of 0 at 1 min and 4 at 5 min. This infant was in the NICU for 133 days with an NMI of 5. This infant falls in the minimal brain injury classification category (1A) and was in the SPEEDI2 late group (receiving intervention from Visit 2 to Visit 3). The mother reported having up to a high school education for herself and the father, and a household income category of USD 15,000 to USD 24,999 (Table 1). The mother reported that the father was working full-time. The infant was discharged home with at least one parent. At NICU discharge this infant had a normal GMA, and a risk of long-term motor impairments was suggested by TIMP scores at Visit 1. Bayley-III scores at Visit 2 indicate superior motor performance (121 composite) and high average cognitive performance (110 composite; Table 2). 

This infant’s TIMP z-score went up (0.284) from Visit 1 to Visit 2, with both visits’ scores suggesting a risk of long-term motor impairments. Over time, this infant steadily improved in their raw scores for cognitive and motor skills; however, both composite scores consistently decreased over time. Cognitive composite scores dropped to 80 composite at Visit 4 and went from the highest to the lowest cognitive composite score of the group from Visit 2 to Visit 4. Motor composite scores also dropped dramatically from 121 composite at Visit 2 to 79 composite at Visit 4. Global problem-solving scores on the APSP increased from Visit 2 to Visit 4. Although this infant improved at each assessment, Infant 1 began with the highest global APSP score at Visit 2 and their slope of change to Visit 3 was much flatter than the other infants in the group (Figure 1). 

Infant 2: Infant 2 was a White biological female born at 27 weeks with a birth weight of 840 g and APGAR scores of 4 at 1 min and 7 at 5 min. This infant was in the NICU for 98 days with an NMI of 5. This infant falls in the minimal brain injury classification category (1A) and was in the SPEEDI2 early group (receiving intervention from Visit 1 to Visit 2). The mother reported having less than a college degree, while a professional school degree was reported for the father, and a household income category of USD 100,000 and over (Table 1). The mother reported that the father was working full-time. The infant was discharged home with at least one parent, sibling, and grandparent. At NICU discharge this infant had a normal GMA, and a risk of long-term motor impairments was suggested by TIMP scores at Visit 1. Bayley-III scores at Visit 2 indicate average motor performance (103 composite) and average cognitive performance (105 composite; Table 2). 

At NICU discharge this infant had the lowest TIMP score of the group at 41 and improved by Visit 2 with a score of 91, and their z-score improved by over half a standard deviation (0.584) from Visit 1 to Visit 2, but the Visit 2 score still suggested a risk of long term motor impairments. Over time, this infant improved in their raw scores for cognitive and motor skills. Cognitive composite scores had a consistent decrease across Visits 2 to 4 and this infant went from 105 composite at Visit 2 to 90 composite at Visit 4. Motor composite scores had a steep drop from 103 composite at Visit 2 to 82 composite at Visit 3, and an increase in composite scores between Visit 3 and 4 from 82 composite to 88 composite. Global problem-solving scores on the APSP increased from Visit 2 to Visit 4, with a rapid increase from Visit 2 to Visit 3, and were the highest global APSP scores in the group at Visit 4 (Figure 1).

Infant 6: Infant 6 was a biological female born at 27 weeks with a birth weight of 1038 g and APGAR scores of 3 at 1 min and 7 at 5 min. This infant was in the NICU for 99 days with an NMI of 5. The infant had a Grade I or II periventricular haemorrhage-intraventricular haemorrhage (PVH-IVH), but the injury was resolved by the time of the first scan so the infant was classified as without brain injury. This infant was in the SPEEDI2 early group (receiving intervention from Visit 1 to Visit 2). Infant’s race, maternal and paternal education levels and employment statuses, household income, infant location post-discharge, and people living with the infant were not reported (Table 1). At NICU discharge this infant had an abnormal GMA, and a risk of long-term motor impairments was suggested by TIMP scores at Visit 1. Bayley-III scores at Visit 2 indicate average motor (103 composite) and cognitive (105 composite) performance (Table 2).

At NICU discharge this infant had a TIMP z-score that went down (−0.304) from Visit 1 to Visit 2, with both visits’ scores suggesting a risk of long-term motor impairments. Although this child had the greatest positive change in their raw score from Visit 1 to Visit 2, they had the lowest z-score in the group at Visit 2 (−1.375). The Bayley-III and APSP were only performed at Visit 2 for this infant so change over time could not be assessed for those measures (Figure 1). 

### 3.3. Case Descriptions: FD, FC, FDC

Similar patterns were observed across the tracts of interest. The left CST was identified as being highly related to overall motor function, so for visualisation, the left CST was chosen to be representative of the change in FD, FC, and FDC in the results; all tracts are presented in the supplementary materials (Appendix A). The left rather than right CST was chosen due to known lateralised asymmetry in white matter development in the first year of life [58]. A summary of each infant’s FD, FC and FDC of the left CST at each visit can be found in Figure 2.

#### 3.3.1. Usual Care Group

Infant 3: Infant 3’s FC and FDC increased at every visit in all tracts. FD increased from Visit 1 to Visit 2 and varied across tracts from Visit 2–4. FDC in the left CST was positively correlated with APSP global scores, and FC was correlated with Bayley-III cognitive raw and composite scores. There were no significant correlations between left CST and any measures and FD (Table 3). 

Infant 4: In all tracts, Infant 4’s FC and FDC increased at every visit, and FD increased from Visit 1 to Visit 2 and then stayed fairly consistent across visits. In Infant 4, Bayley-III raw cognitive scores positively correlated with FDC in the left CST. FC in the left CST was positively correlated with APSP global scores. There were no significant correlations between left CST and any measures and FD (Table 3).

Infant 5: In all tracts, Infant 5’s FD, FC, and FDC in the left CST increased from Visit 1 to Visit 3, this baby did not have an MRI at Visit 2 or 4. No correlations could be calculated for this infant due to the limited number of MRI scans.

#### 3.3.2. SPEEDI2 Intervention Group

Infant 1: In all tracts except R ICP, Infant 1’s FC and FDC in the left CST increased at every visit. FD increased from Visit 1 to Visit 2 then stayed fairly consistent across visits in all tracts except CC-2. In Infant 1, across visits, FDC in the left CST was positively correlated with both Bayley-III raw gross motor scores and APSP global scores. FC in the left CST was positively correlated with Bayley-III cognitive raw scores. There were no significant correlations between left CST and any measures and FD (Table 3).

Infant 2: In all tracts Infant 2’s FC and FDC increased at every visit, and FD increased from Visit 1 to Visit 2 and then stayed fairly consistent across visits. FC in the left CST was positively correlated with Bayley-III cognitive raw scores and the APSP global scores. There were no significant correlations between left CST and any measures and FD (Table 3).

Infant 6: Infant 6 only received MRI at Visits 1 and 2. This child’s FD, FC, and FDC in all tracts increased from Visit 1 to Visit 2. No correlations could be calculated for this infant due to the limited number of MRI scans.

### 3.4. Group Summary

Every child improved in Bayley-III Cognitive and Gross Motor raw scores, in APSP scores from Visit 2 to Visit 4, and in the raw scores of the TIMP from Visit 1 to Visit 2 indicating they were all learning new skills. Significant positive change was seen in Bayley-III cognitive (*p* < 0.001) and gross motor (*p* < 0.001) raw scores and APSP global problem-solving (*p* = 0.005) scores from Visit 2 to Visit 4. However, on Bayley-III composite scores, group means decreased from Visit 2 to Visit 4, but these changes were not significant (Cognitive, *p* = 0.382; Motor, *p* = 0.086). These results reflect that all children are improving in their cognitive and motor skills over time (raw), but as a group, they are not improving at a rate that is consistent with their age-matched peers (composite/norm-referenced).

The pattern of brain injury categories stayed the same across time for each infant across all visits, and upon visual inspection, no obvious pattern in mean FC, FD or FDC emerged when comparing those with and without brain injury. Mean FD, FC, and FDC across all participants significantly increased from Visit 1 to Visit 3 in all tracts (*p*’s < 0.005). As a group, and within each individual, the largest increase in FD, FC, and FDC of all tracts occurred from Visit 1 to Visit 2.

In the left CST, the greatest change in FC and FDC from Visit 1 to Visit 2 was seen in Infant 2, who made gains on the TIMP z-score of more than 0.5 standard deviations and received the SPEEDI2 early intervention during the same time period. The greatest change from Visit 2 to Visit 3 was seen in Infant 4 (usual care) for FDC, Infant 2 (SPEEDI2 early) for FC, and Infant 1 (SPEEDI2 late) for FD. The greatest change from Visit 3 to Visit 4 occurred in Infant 4 (usual care) for FD, and Infant 1 (SPEEDI2 late) for FC and FDC. The two infants with the highest FDC of the left CST across all visits were both White non-Hispanic infants, with at least one parent with a professional degree, and a parent-reported annual income category of USD 100,000 or over. Importantly, even for two of the infants with increased developmental risk, Infant 1 (extremely preterm, SPEEDI2 late) and Infant 2 (low birth weight, SPEEDI2 early), we see positive change over time in imaging metrics and outcome measures.

## 4. Discussion

Diffusion MRI-derived metrics and developmental outcomes varied among the 6 infants in this case series, highlighting the importance of considering individual characteristics, prediction of developmental delay and the implications for brain change in relation to intervention and adaptive development. In preterm infants, consistent with previous literature we show that changes in white matter microstructure and macrostructure are positively correlated with cognitive and motor development over time [18,20], and secondly, that they are positively associated with better outcomes on a convenient play-based measure of motor problem-solving development; which has not been not previously observed in any infant populations. We find that in all infants, the greatest increase in FD, FC, and FDC occurs between Visit 1 and 2 (mean chronological age: 2.68–6.22 months), a phase of rapid axonal growth and myelination. This suggests that this is a potential window of time to optimally support adaptive development. Positive change in developmental assessment and imaging metrics was observed in infants receiving an intervention, supporting previous literature suggesting that early physical therapy intervention may be beneficial to development [24,59,60,61]. This result warrants further studies with larger groups to formally compare the impact of an intervention to usual care. Across all participants, regardless of assignment in SPEEDI intervention or usual care grouping, we observed the greatest FDC of left CST in two white children born at 27 weeks with the highest SES, and a parent with a professional degree, despite both being in the brain injury group, both having a TIMP score indicating high risk for long term motor impairments at visit 2, and one having sepsis, demonstrating that socio-demographic and socio-economic factors may play a role in developmental and neural outcomes in infants born preterm. While this small sample size prohibits any causative inference, individual case results may support calls for targeted efforts to reduce disparity in access to high-quality, early, personalised therapy [62,63,64].

## 5. Limitations and Future Directions

Although our results support neuroplastic change in very preterm infants with and without intervention, pertinent conclusions about intervention-induced longitudinal changes cannot be drawn from such a small sample size. Another limitation in group comparison is that the infants in the intervention group did not all receive intervention during the same time period. To compare the timing of intervention, the primary study included an “early” and “late” intervention group. Further, although caregivers reported they were adjusting well to the intervention and using intervention strategies at home, there may be variability in caregiver understanding and adherence to the intervention. High-quality randomised controlled designs are needed to interpret the neuroplastic changes in response to dose and timing of intervention using larger sample sizes. Measuring the impact and timing of intervention longitudinally would contribute to our understanding of which aspects of the intervention are most important and effective. These data are crucial for targeted and personalised approaches to rehabilitation.

## 6. Conclusions

In this case, a series of six infants born VPT, we investigated whether changes in FD, FC, and FDC in regions associated with motor learning were associated with motor, cognitive, and motor-based problem-solving outcomes and if early intervention can impact these relationships over the first year of life. We demonstrated that in this sample, changes in white matter microstructure and macrostructure are positively correlated with cognitive and motor development over time. Interventions such as SPEEDI may be beneficial to behavioural and neural development. The results of this study support the use of physical therapy provided through a parent–infant therapist collaborative with key intervention ingredients that define this novel intervention approach. While the sample size prohibits causative inference, clinicians can use the information in this paper to focus on the need for well-timed, activities that challenge the infant’s motor and thinking skills at the same time. Understanding change in diffusion MRI-derived metrics, their association with development, and change induced by intervention, can facilitate the application of targeted early interventions during a period of neuroplasticity in high-risk infants.

## Figures and Tables

**Figure 1 jpm-12-02024-f001:**
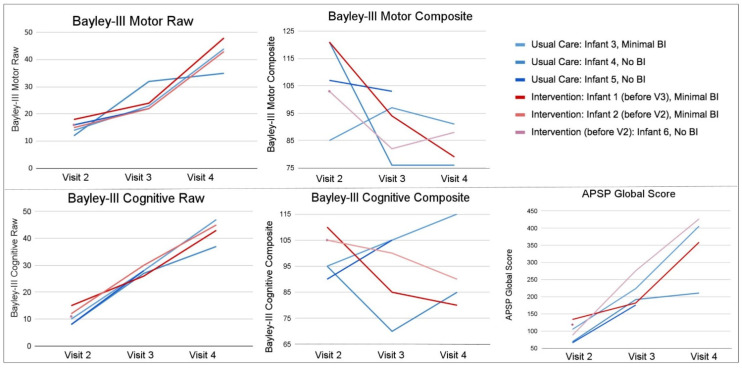
Outcome Measures at Each Visit. Figure 1 shows individual infant scores at each visit for motor, cognitive and motor-based problem solving outcomes. Infant 6 is not included due to lack of Visit 2 data. Usual care infants are coded blue and intervention infants are coded red. Bayley-III = The Bayley Scales of Infant and Toddler Development, 3rd edition, APSP = Assessment of Problem Solving in Play. Raw scores represent the number of points earned so that an increase in raw score can be interpreted as the child completing more items. The composite scores are a comparison with the normative same, have mean of 100 and standard deviation of 15. Thus, a composite score that does not change indicates the child is learning at the same rate as the normative data, in that domain. An increase in composite scores reflects gaining skills at a faster rate than the normative sample. A decreasing composite score reflects that the child is not gaining skills at the same rate as the normative sample but must be considered along with the raw score to determine if the child is gaining new skills slowly or not at all.

**Figure 2 jpm-12-02024-f002:**
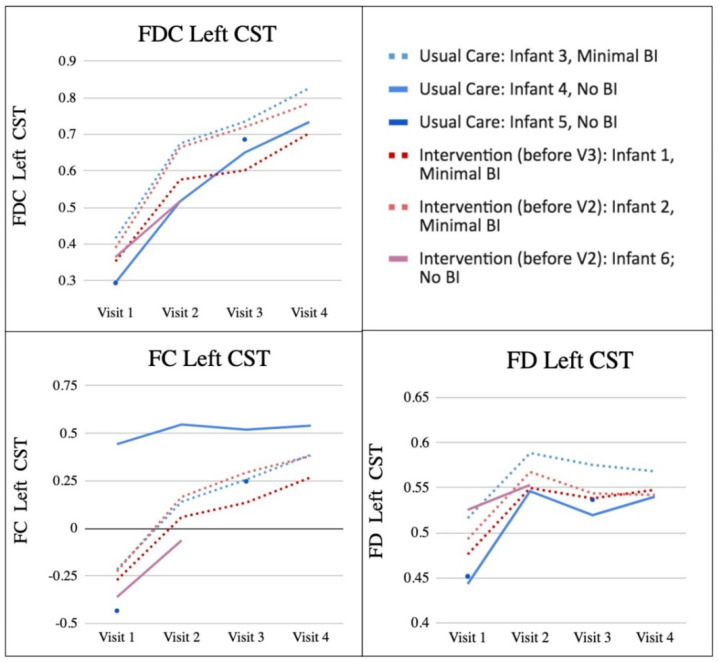
FDC, FD and FC at Each Visit. Figure 2 shows individual infant metrics of FDC, FC, and FD at each visit (these are all expressed in arbitrary units). Usual care infants are coded blue and intervention infants are coded red. Infants with minimal brain injury have dashed lines, and infants with no brain injury have solid lines. BI = brain injury, FD = fibre density, FC = fibre bundle cross-section, FDC = combined fibre density and bundle cross-section, CST = corticospinal tract.

**Table 1 jpm-12-02024-t001:** Baseline information.

Variable	Group	Brain Injury	PVH-IVH	Sepsis	Baseline GMA	TIMP Z-Score	Gestational Age	Birth Weight	Gender	Race; Ethnicity	Annual Income
Infant 1	SPEEDI	Minimal	No	No	Normal	−0.88	23 weeks	580 gm	M	Black; Non-Hispanic	USD 15K–USD 25K
Infant 2	SPEEDI	Minimal	No	No	Normal	−1.36	27 weeks	840 gm	F	White; Non-Hispanic	≥USD 100K
Infant 3	Usual Care	Minimal	No	Yes	Normal	−0.08	27 weeks	1030 gm	M	White; Non-Hispanic	≥USD 100K
Infant 4	Usual Care	Normal	No	No	Normal	0.15	27 weeks	970 gm	F	Black; Non-Hispanic	USD 35K–USD 49K
Infant 5	Usual Care	Normal	No	No	Normal	0.8	27 weeks	1000 gm	F	Not Reported	Not Reported
Infant 6	SPEEDI	Normal	Grade I or II	No	Abnormal	−1.07	27 weeks	1038 gm	F	Not Reported	Not Reported

Note: SPEEDI = Supporting Play Early Exploration and Developmental Intervention, M = Male, F = Female, gm = grams, K = thousand, TIMP = Test of Infant Motor Performance at baseline, GMA = General Movement Assessment, PVH-IVH = Periventricular/Intraventricular Haemorrhage.

**Table 2 jpm-12-02024-t002:** Behavioural outcomes.

Variable	Age *	TIMP Z-Score	GMA	Bayley-III MC	Bayley-III MC	Bayley-III MC	Bayley-III CC	Bayley-III CC	Bayley-III CC
	Visit 1; Visit 2; Visit 3; Visit 4	Visit 2	Visit 2	Visit 2	Visit 3	Visit 4	Visit 2	Visit 3	Visit 4
Infant 1	4m 7d; 7m 14d; 10m 6d;17m 25d	−0.59	Normal	121	94	79	110	85	80
Infant 2	2m 17d; 6m 1d; 10m 2d; 16m 11d	−0.77	Normal	103	82	88	105	100	90
Infant 3	2m 9d; 5m 26d; 8m 26d; 16m 14d	−0.73	Normal	85	97	91	95	105	115
Infant 4	2m 4d; 5m 14d; 13m 12d; 15m 22d	0.38	Normal	121	76	76	95	70	85
Infant 5	1m 26d; 5m 22d; 9m 5d; 20m 23d	−0.36	Normal	107	103	94	90	105	85
Infant 6	3m; 6m 23d	−1.38	Normal	103	N/A	N/A	105	N/A	N/A

Note: * chronological age, m = months, d = days, TIMP = Test of Infant Motor Performance, GMA = General Movement Assessment, Bayley-III MC = The Bayley Scales of Infant and Toddler Development 3rd edition Motor Composite; Bayley-III CC = The Bayley Scales of Infant and Toddler Development 3rd edition Cognitive Composite.

**Table 3 jpm-12-02024-t003:** Correlations with FD, FC, FDC.

Participant	Variable	Bayley-III CR		Bayley-III CC		Bayley-III GMR		Bayley-III MC		APSP Global Score	
		r	*p*	r	*p*	r	*p*	r	*p*	r	*p*
Infant 1 (Intervention, minimal injury)	FD Left CST	−0.047	0.97	0.512	0.658	0.163	0.896	0.327	0.788	0.149	0.904
	FC Left CST	0.999 *	0.021	−0.866	0.333	0.984	0.113	−0.949	0.203	0.987	0.104
	FDC Left CST	0.978	0.135	−0.763	0.447	1.00 **	<0.001	−0.878	0.317	1.00 **	0.01
Infant 2 (Intervention, minimal injury)	FD Left CST	−0.917	0.261	0.795	0.415	−0.737	0.473	0.942	0.218	−0.921	0.254
	FC Left CST	0.99 *	0.037	−0.955	0.191	0.924	0.249	−0.768	0.442	0.999 *	0.030
	FDC Left CST	0.995	0.063	−0.99	0.091	0.973	0.149	−0.659	0.542	0.994	0.07
Infant 3 (Usual Care, minimal injury)	FD Left CST	−0.983	0.118	−0.986	0.108	−0.922	0.252	−0.639	0.559	−0.958	0.184
	FC Left CST	1.00 *	0.012	0.999 *	0.02 *	0.981	0.123	0.47	0.688	0.996	0.055
	FDC Left CST	0.994	0.067	0.993	0.077	0.994	0.067	0.392	0.744	1.00 **	<0.001
Infant 4 (Usual Care, no injury)	FD Left CST	−0.388	0.747	0.982	0.12	−0.584	0.603	0.678	0.526	−0.582	0.685
	FC Left CST	0.989	0.093	−0.688	0.534	0.997	0.051	−0.98	0.128	0.997 *	0.049
	FDC Left CST	0.999 *	0.02*	−0.515	0.656	0.964	0.172	−0.924	0.249	0.964	0.17
Infant 5 (Usual Care, no injury)	FD Left CST										
	FC Left CST										
	FDC Left CST										
Infant 6 (Intervention, no injury)	FD Left CST										
	FC Left CST										
	FDC Left CST										

Note: Bayley-III = The Bayley Scales of Infant and Toddler Development, 3rd edition, APSP = Assessment of Problem Solving in Play, CR = cognitive raw, CC = cognitive composite, GMR = gross motor raw, MC = motor composite, FD = fibre density, FC = fibre cross-section, FDC = fibre density and cross-section, CST = corticospinal tract, * *p* < 0.05, ** *p* < 0.01.

## Data Availability

Not applicable.

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
