# Peer review of "Effect of a NICU to Home Physical Therapy Intervention on White Matter Trajectories, Motor Skills, and Problem-Solving Skills of Infants Born Very Preterm: A Case Series"

_jpm, 2022, doi:10.3390/jpm12122024_

Round 1

Reviewer 1 Report

This paper presents important preliminary evidence via case series design for a parent-mediated, play-based developmental intervention for pre-term infants who are at-risk for long-term motor and developmental delays/conditions.

The method section would benefit from a more thorough description of the SPEEDI2, rather than referencing the protocol paper. 

Limitations of a small sample and case-series design were clearly discussed, however additional details of potential confounding factors such as caregiver's understanding of the intervention and their capacity for adherence to the protocol during the intervention could be included.

If there are plans for a long-term follow-up of these infants, mention of this would add value to the paper.

Author Response

We thank the reviewer for their helpful comments. The methods section has been updated to include more information about the SPEEDI2 intervention. The limitations section has also been updated to include the caregivers understanding of the intervention. 

Reviewer 2 Report

This is a well-written manuscript that took a lot of time and effort. Incorporating sociodemographic data while assessing neurodevelopment progress provides essential insight that is often overlooked. Emphasis on parents' educational background, involvement in childcare, and social demographics in your manuscript is a bonus. As you already commented in the manuscript, it is a very small sample size. Results are difficult to interpret since 5/6 babies were 27 weeks at birth, whereas 1/6 was 23 weeks and in the periviable category. Additionally, the intervention group includes the 23-weeker and a second 27-weeker with abnormal general movement assessment at baseline. 

Specifics such as intraventricular hemorrhage, hypo or hyperglycemia, severe electrolyte imbalances, sepsis, sedation, and steroid use would be helpful in addition to time to attain full per oral feeds. 
In conclusion, adding your recommendations regarding the timing of initiating specific interventions would be helpful. 

Author Response

We thank the reviewer for their helpful comments. The participant description tables have been updated to include available additional details as recommended. We did not have access to some of the requested data, and acquiring it would require going back into the medical record data. We choose not to do this because the remaining requested data is not immediately pertinent to the primary questions of the paper.  Our recommendations regarding timing of early intervention have been added to the conclusions.

Reviewer 3 Report

What are the limitations of this work? 

Are there any statistical methods used? 

How does this compare with other literature studies? 

Author Response

What are the limitations of this work? 

Please see the “limitations” section, pasted here for convenience. “Although our results support neuroplastic change in very preterm infants with and without intervention, pertinent conclusions about intervention induced longitudinal changes cannot be drawn from such a small sample size. Another limitation in group comparison is that the infants in the intervention group did not all receive intervention during the same time period. To compare timing of intervention, the primary study included an “early” and “late” intervention group. Further, although caregivers reported they were adjusting well to the intervention and using intervention strategies at home, there may be variability in caregiver understanding and adherence to intervention.”

Are there any statistical methods used? 

Please see the “statistical analysis” section, pasted here for convenience. “Diffusion MRI derived metrics (FD, FC, FDC). Blinded team members performed typical state-of-the-art fixel-based analysis pipeline steps [36-37], including denoising [50], Gibbs-ringing correction [51], motion correction [52], estimation of 3-tissue response functions and averaging response functions across the images from the fourth (oldest) timepoint [53], upsampling to 1.5 mm isotropic voxels, brain mask estimation [54], Single-Shell 3-Tissue Constrained Spherical Deconvolution [40], and intensity normalisation (global and bias fields) [55]. These steps resulted in white matter fibre orientation distributions (FODs) and grey matter and CSF compartments for each participant. A study-specific white matter FOD template was constructed, and all infants’ FOD images were registered and warped to this template [56]. In template space, fixels were segmented and reoriented, correspondence with common template fixels was established, and fixelwise fibre density (FD), fibre cross-section (FC, in log form) and fibre density and cross-section (FDC) metrics were computed for each infant, as previously described [36]. White matter tracts were segmented in template space using TractSeg [57], following which FD, FC and FDC values for each infant were averaged across each tract, plotted, and a Hedges g calculated. Note these metrics are entirely tract-specific, and not influenced by those of other crossing white matter tracts. 

General. Descriptive statistics were computed in IBM SPSS Statistics Edition 28 and were used to describe the demographic details of the participants. Within and across all infants, Pearson correlation and paired samples t-tests were used to analyze change over time. Due to the case series design and small sample size, a complete statistical group analysis was beyond the scope of this project. Qualitative and descriptive changes for developmental measures, FD, FC and FDC, and brain injury classification at Visit 1, Visit 2, Visit 3, and Visit 4 were reported. To explore the role of intervention on neuroplastic changes, FD, FC, and FDC, and brain injury classification for all infants who had at least a baseline and 12 months visit from each group (SPEEDI2 intervention [collapsed] and usual care), were visually represented using line graphs and descriptively analyzed.”

How does this compare with other literature studies?

As mentioned in the discussion, in agreement with previous studies in preterm infants, we show that changes in white matter microstructure and macrostructure are positively correlated with cognitive and motor development over time [18, 20]. Positive change in developmental assessment and imaging metrics was observed in infants receiving intervention, supporting previous literature suggesting that early physical therapy intervention may be beneficial to development [59, 24, 60-61]. We observe the greatest FDC of left CST in white children with the highest SES, and a parent with a professional degree, demonstrating that socio-demographic and socioeconomic factors may play an important role in responsiveness to early intervention, and supports calls for targeted efforts to reduce disparity in access to high quality, early, personalized therapy [62-64].